# Suction Influence on Load–Settlement Curves Predicted by DMT in a Collapsible Sandy Soil

**DOI:** 10.3390/s23031429

**Published:** 2023-01-27

**Authors:** Alfredo Lopes Saab, André Luís de Carvalho Rodrigues, Breno Padovezi Rocha, Roger Augusto Rodrigues, Heraldo Luiz Giacheti

**Affiliations:** 1Department of Civil and Environmental Engineering, São Paulo State University, Bauru 17033-360, SP, Brazil; 2Advanced Campus of Ilha Solteira, Federal Institute of São Paulo, Ilha Solteira 15385-000, SP, Brazil

**Keywords:** load–settlement curve, plate load test, DMT, unsaturated soil, inundation

## Abstract

The plate load test (PLT) is the most reliable in situ testing for studying the load–settlement behaviour of footings on unsaturated collapsible soils. In these soils, the suction profile is not constant along the depth, and the scale effect between the prototype and footing leads to different suction averages and, consequently, different data. One method to eliminate the effect of soil suction on the test data is to fully saturate the soil prior to the test, which is also recommended at the design process for footing on collapsible soils. However, the inundation process on PLTs is expensive and time-consuming, which makes this procedure difficult to incorporate into engineering practice. This study presents a device that can be attached to flat dilatometer (DMT) to allow local inundation of the soil as part of the in situ test campaign and obtain the DMT-constrained modulus (*M_DMT_*) for both natural and inundated conditions. The *M_DMT_* presented an average reduction of 56% from natural to inundated condition. This parameter can be used in a model to predict load–settlement curves by DMT data considering the suction influence on this behaviour. The curves obtained from the prediction model were compared to curves determined by PLT conducted under the same in situ conditions. Good agreement was found between the curves predicted by DMT and those measured by PLT for both conditions. The proposed procedure, which uses a device attached to the DMT blade, provides an investigation method to obtain the load–settlement curve under different suction conditions, which can help in the selection and performance prediction of shallow foundations, taking into account suction and collapse phenomenon-related problems.

## 1. Introduction

The prediction of bearing capacity and settlements in shallow foundation design is essential for good construction performance. Currently, shallow foundation design approach focuses more on settlement criterion than bearing capacity criterion [1]. Several methods that aim to predict the load-settlement curve by means of in situ and laboratory tests have been developed [2,3,4]. The settlement prediction is a function of analytical models and soil stiffness [5]. For sandy soils, in situ tests (e.g., standard penetration test—SPT [6]; cone penetration test—CPT [2]; pressuremeter test—PMT [4]; flat dilatometer—DMT [7,8]) have been widely used to determined soil stiffness due to the difficulties in collecting undisturbed soils samples and thus predict the settlements in shallow foundation. It should be mentioned that most of the correlations available in the literature to estimate soil stiffness are empirical and based on in situ test data from tests such as CPT and SPT [9].

According to Marchetti et al. [10], predicting settlements of shallow foundations is probably the main application of the DMT, especially in sands, where undisturbed sampling and estimating compressibility are particularly difficult. DMT allows for a quick and appropriate settlement estimate for foundations installed in several soil types [8]. The settlement estimate is meant to be the settlement under the “working conditions,” i.e., for a safety factor *Fs* ≈ 2.5–3.5 [7,10]. However, only a few studies have assessed the potential of this technique for predicting the entire load-settlement curve [9,11].

The prediction of the load-settlement curves of shallow foundations based on DMT was studied by Dos Santos et al. [9] and Silva et al. [11]. The authors used Décourt’s [1] proposal for predicting load–settlement curves under both natural and inundated conditions based on DMT data, as well as to evaluate the effect of inundation on the bearing capacity and the settlements of plate load tests on collapsible soil.

Collapsible soils are usually unsaturated, have a metastable structure and low density and are susceptible to a sudden decrease in total volume or collapse upon wetting, causing problems such as differential settlements, as well as risks of slope ruptures [12,13,14,15].

Techniques have been developed to treat collapsible soils, such as compaction [16,17], chemical stabilization [18,19], soil pre-wetting [16], and soil reinforced with polymeric strips [20]. However, foundations of engineering works are often supported by collapsible soils in their natural state.

The soil collapse mechanism is due to the metastable structure of the soil, which in the unsaturated condition is kept in equilibrium due to temporary bonds (i.e., surface tension and soluble cements). Inundation reduces the soil suction by dissipating the temporary bonds, reducing the contact pressure between soil particles, and dissolving the soil cementation, which causes the collapse of its structure.

Soil collapsibility can be investigated in the laboratory by means of single or double oedometer tests [21,22,23] or in situ by means of load tests on footings or plates [24] under natural and inundated conditions.

It is necessary to determine the collapse stress (*σ_c_*) for the design of shallow foundations on collapsible soils, which represents the minimum stress that causes collapse in a collapsible soil, in addition to the admissible settlement for the inundated soil condition (i.e., the most critical condition) [25]. For this in situ determination, load tests are carried out on plates installed in a pit prepared to maintain, around the plate, water with sufficient flow to guarantee a water column for a period of 48 h prior to the test, as well as during the stages of the load test [25]. The process of soil inundation reduces the soil suction along all the investigated soil profiles, which also reduces the bearing capacity of the collapsible soil. This behaviour was observed for both shallow [24] and deep [26] foundations. The reduction in bearing capacity due to soil inundation can reach approximately 70% [26].

Researchers noticed the relevance of the soil suction on load–settlement curves [24,27,28], especially on collapsible soils [24]. Costa et al. [24] investigated the effect of soil suction in a collapsible tropical sandy soil by means of plate load tests and noticed that a small increase in suction from 0 to 10 kPa significantly influenced the load-settlement curve and led to an increase of approximately 100% in the failure stress of the soil–plate system tested.

Cintra and Aoki [25] discuss the importance of incorporating load and penetration tests in natural and inundated conditions into engineering practice to assess the mechanical behaviour of collapsible soils. However, the inundation process is expensive, time-consuming and difficult to implement as a current practice for site investigation campaigns. In addition, the inundation process is difficult to perform for deeper soil layers.

This study presents and discusses the comparison between the load–settlement curves obtained by PLT on a rigid plate with 300 mm diameter and predicted from DMT for a collapsible tropical sandy soil. Comparisons were performed for both natural and inundated conditions. The predictions for inundated curves from DMT were performed based on *M_DMT_* obtained from a device attached to the DMT that allows local inundation. This approach permits the determination of DMT soil parameters in the natural and inundated conditions along the soil profile and the assessment of soil collapsibility.

## 2. Flat Dilatometer Tests (DMT)

### 2.1. Test Description and Interpretation

The interpretation of flat dilatometer test (DMT) data is based on empirical correlations with the test pressure readings, resulting in three intermediate parameters: the material index (*I_D_*), horizontal stress index (*K_D_*) and dilatometer modulus (*E_D_*), expressed by Equations (1)–(3), respectively. The modulus *M* determined from DMT (often designated *M_DMT_*) is the vertical drained confined (one-dimensional) tangent modulus at effective vertical stress (*σ′_v_*_0_) and is the same modulus obtained by oedometer [10]. *M_DMT_* can be obtained by Equation (4). This modulus depends on a correction factor (*R_M_*) (see Table 1), which is a function of the material index (*I_D_*) and horizontal stress index (*K_D_*).
(1)$$I_{D}=\frac{p_{1}-p_{0}}{p_{0}-u_{0}}$$
(2)$$K_{D}=\frac{p_{0}-u_{0}}{\sigma \prime_{v}}$$
(3)$$E_{D}=34.7\left(p_{1}-p_{0}\right)$$
(4)$$M_{DMT}=R_{M}E_{D}$$
where *p*_0_ is the lift-off pressure, *p*_1_ is the expansion pressure, *u*_0_ is the hydrostatic water pressure and *σ′_v_* is the vertical effective stress.

### 2.2. Load–Settlement Curve Prediction from DMT

Décourt [1] analysed 145 load tests on footings and plates and proposed an approach to interpret the load–settlement curve based on the normalization of stresses by the conventional bearing capacity and the settlements by the equivalent width (*B_eq_* is the square root of the base area) of the foundation or plate. The author states that there is no influence of the scale effect, depth, or geometry on the normalized load–settlement curve. Décourt [1] considers the conventional failure stress (*q_uc_*), which corresponds to a settlement of 10% of the plate equivalent width (*B_eq_*) as the bearing capacity parameter of Terzaghi [29].

Dos Santos et al. [9] and Silva et al. [11] presented a procedure to predict the load–settlement curve by means of DMT results based on the Décourt [1] approach. Dos Santos et al. [9] used the Lehane and Fahey [30] considerations about the operational modulus, or working modulus (*M_DV_*), to predict a complete load–settlement curve. However, *M_DV_* is little affected by seasonal variability (i.e., soil suction), as discussed by Silva et al. [11]. Then, in the following sections, the prediction of the load–settlement curves for the inundated and natural conditions was made by using the average *M_DMT_* values within the zone of influence of the foundation element, considered equal to 2*B* (*B* = 0.30 m). The load–settlement curve can be obtained from the DMT results data in eight steps:Use the DMT results data and empirical correlations to obtain the intermediate parameters along the depth;Calculate the constrained modulus (*M_DMT_*) along the zone of influence of the plate, plausibly adopted equal to 2*B* (two times the plate width);Calculate the stress applied to the footing (*q_app_*) by Equation (5):
(5)$$q_{app}=4\;.\;\left(\frac{s}{B}\right)\;.\left(\frac{M_{DMT}}{\pi }\right)$$
where *s* is the settlement and *B* is the plate width;

4.Assume *C* = 0.42, according to Décourt [31];5.For the DMT working condition (*s*/*B* = 1.8%), assume the relation *q_app_*/*q_uc_* is equal to 0.486;6.Calculate *q_uc_* from the Steps 3 and 5;7.In the case of circular foundation or plate, the equivalent width (*B_eq_*) should be determined by the square root of the base area of such foundation or plate, as recommended by Décourt [1];8.Obtain the load–settlement curve by Equation (6):


(6)
$$\log\frac{q}{q_{uc}}=C+C.\log\frac{s}{B_{eq}}$$


### 2.3. DMT after Inundation

Rodrigues [32] proposed a procedure that uses a device attached to the DMT blade that enables local inundation prior to the test. This device consists of two parts: a male-female thread (Part 1) that is connected to the DMT rods and a hollow rod with female-female thread (Part 2) that is connected to the first part and the DMT blade. Figure 1 shows the parts of the device and the system attached to the DMT blade. Tests were carried out in the laboratory and in situ to establish the best way to promote soil inundation through Part 1 of this device. A hollow steel ring with two flat o-rings on the upper and lower parts was used. A nonwoven geotextile (type GeoFort 100% propylene GR 200–2200 g/m—grey—from Ober S/A) was used to avoid obstruction of the steel device pores. 

A water pump injection (12 V and 80 W) was used to guarantee a constant water flow during the test imposing a constant flow of approximately 0.55 L/min during the soil inundation process. A 4 mm diameter water hose was used to conduct water from reservoir to the inundation device. Volumes of 25, 50, 100, and 200 L of water were injected into the soil to select an efficient inundation procedure prior to DMT testing. Soil samples were collected using a helical auger to determine the post-inundation moisture content of the soil [33] after each inundation process to assess the influence zone and its effect on the DMT readings as a function of the volume of water used in the inundation. Rodrigues [32] suggested using a volume of 100 L because it provided an influenced zone reaching 0.60 m below the inundation point. It is important to note that depending on the flow rate of the pump, the injection of 100 L of water into the soil takes approximately 3 h. Figure 2 presents a schematic representation of the DMT with the inundation device.

The DMT was carried out under natural soil conditions (Figure 3a), every 0.20 m depth intervals, up to reaching the depth of interest (Figure 3b). Therefore, the soil is locally inundated with 100 L of water (Figure 3c). After inundation, the DMT blade is pushed into the soil, and DMT readings are carried out at 0.20 m depth intervals (Figure 3d) up to reaching the next depth of interest where the inundation process is repeated. As the zone influenced by the local inundation reaches 0.60 m below the depth of interest, after each inundation stage, the following three sets of readings represent the DMT data for the inundated soil condition. Therefore, as the zone of influence for shallow foundation is equal 2*B*, the average DMT parameters calculated from these readings can be assumed to represent the load–settlement curve for the plate of 300 mm diameter (i.e., zone of influence equal to 0.60 m) for inundated soil conditions.

## 3. Plate Load Tests (PLT)

Oh and Vanapalli [34] discuss the effect of soil suction and geometry on PLT on unsaturated soils. The authors pointed out that in the case of unsaturated soils, the zone of influence of the plate and the suction distribution profile must be considered, as different widths lead to different average suction values below the plate and, consequently, different results. Therefore, compatible influence zones must be defined to compare load–settlement curves determined by different tests. The zone influenced by the local inundation procedure using the device attached to the DMT is approximately 0.60 m, as previously presented. Therefore, since the curve predicted from DMT is obtained by the average parameters at this depth, it is important to compare the predicted curves to curves determined by prototypes with an adequate scale (i.e., zone of influence equivalent to 0.60 m). Therefore, load tests were carried out on plates of 300 mm diameter (Figure 4), assuming a zone of influence equal to 2*B* = 0.60 m. The plate load was the slow maintained load (SML) [35]. The tests were carried out along the active zone [36] of the experimental research site at Unesp, in the city of Bauru, inland of São Paulo State, Brazil, at depths of 1.0, 2.0, 3.0, and 4.0 m by means of two distinct campaigns: (a) under natural soil conditions and (b) after inundating the plate test with a 50 mm water column at the base of the pit for a period of 24 h prior to starting the test (Figure 5).

The anchoring rods of the reaction system were installed into the soil by a semiautomatic device of a multipurpose system (TG 63-150 from Pagani Geotechnical Equipment). The main advantage over conventional reaction systems is that no permanent structures, such as reaction piles, are needed. The anchoring rods are removed and can be reused in the assembly of the other plate load tests.

## 4. Study Site

PLT and DMT were conducted at the experimental research site of São Paulo State University (Unesp), located in the city of Bauru, State of São Paulo, Brazil. According to De Mio [37], the region of the study site belongs to the Paraná Sedimentary Basin and is inserted in the western plateau, which is formed by rocks from the Bauru group (Marília and Adamantina Formations), covering the volcanic rocks (basalts) of the Serra Geral Formation that crop out towards the Tietê River valley.

The soil profile at the study site is a red clayey fine to medium sand with high porosity, whose relative density increases with depth. It is classified as an SM by the unified soil classification system (SUCS). The upper horizon (up to approximately 13.0 m depth) is a colluvial Neo-Cenozoic deposit with lateritic behaviour (LA’) overlying a residual soil derived from the weathering of sandstone. The groundwater level was not found until up to 30 m depth. These soils have undergone paedogenic and morphogenetic processes, which typically take place in tropical zones, resulting in partly saturated high-permeability (10^−5^ to 10^−6^ m/s) soils with cohesive–frictional behaviour. Figure 6 shows the grain size distribution, some index properties, cone penetration (CPT), and standard penetration (SPT) test data along the soil profile of the study site.

The climate of the city of Bauru is classified as tropical according to the classification of Köppen–Geiger. Seasonal variability is expected throughout the year for this type of climate, with dry winters (May to September) and rainy summers (December to March). This leads the site condition to changes in moisture content and consequently in soil suction. This suction variability influences the mechanical behaviour of the soil, as can be verified by means of laboratory tests [40] and in situ tests [36,41,42]. Therefore, for unsaturated soils, suction monitoring is necessary. Figure 7 shows suction data obtained by granular matrix sensors (GMSs) at depths of 0.4, 0.9, 2.0, 3.0, 4.0, and 5.0 m at the study site. Overall, it was noted that the suction at depths of 3.0, 4.0, and 5.0 m were less affected by precipitation than at the ground surface. However, there were sudden reductions in suction after long periods of precipitation, as observed in the period between the end of 2016 and the beginning of 2017. The maximum peak of suction is approximately 140 kPa for 2.0 m depth, a value slightly lower than the peaks recorded at depths of 0.4 and 0.9 m, approximately 200 kPa, which is the limit of the equipment used.

This seasonal variability can be verified from DMT data. Figure 8 presents the intermediate DMT parameters from twelve tests in terms of *I_D_*, *K_D_*, and *E_D_* data plotted as an average profile as well as plus and minus one standard deviation (*SD*) obtained from natural conditions. As discussed previously, unsaturated soils in tropical climatic regions experience seasonal changes in water content and, consequently, in soil suction due to wetting and drying cycles, as shown in Figure 7. It may influence soil behaviour and consequently the intermediate DMT parameters (*I_D_*, *K_D_*, and *E_D_*).

## 5. Results and Discussion

### 5.1. DMT

In this study, fifteen DMTs were analysed: twelve were carried out under natural conditions (Figure 8), and three were carried out under inundated conditions (Figure 9). Figure 9 presents the intermediate DMT parameters in terms of *I_D_*, *K_D_*, and *E_D_* obtained from DMT with local inundation as well as the influenced zone (i.e., approximately 0.60 m) (Table 2). The local inundation procedure was carried out at depths of 1.1, 2.1, 3.1, and 4.1 m. This figure shows that the inundation process caused a reduction in *K_D_* and *E_D_* and little change in *I_D_* along the zone influenced by the local inundation. This behaviour can be observed when compared with the average *I_D_*, *K_D_*, and *E_D_* profiles determined under natural conditions. The reduction in *K_D_* and *E_D_* and little change in *I_D_* due to inundation were also observed and discussed by Lutenegger [43]. Changes in the DMT intermediate parameters have a direct effect on the estimate of the deformability soil parameters, such as the constrained modulus (*M_DMT_*), and consequently on the prediction of the complete load–settlement curve.

### 5.2. Constrained Modulus

One of the main applications of the DMT is to predict settlements of shallow foundations [7,10]. Monaco et al. [7] presented a compilation of documented case histories, including comparisons of DMT-calculated vs. observed settlements, to evaluate the accuracy of settlement predictions based on DMT. The authors observed that the average ratio between estimated and measured settlements was approximately 1.3. This fact is related to the good agreement between the constrained modulus obtained by DMT (*M_DMT_*) and other techniques, such as plate load and oedometer tests [10].

In this study, the constrained modulus (*M*) was determined based on DMT, oedometer and plate load tests carried out under natural and inundated conditions. The constrained modulus of the plate load test (*M_PLT_*) was calculated to a settlement ratio of 1.8%, equivalent to that of the DMT, based on previously presented Equation (5).

The prediction of the load–settlement curves for the inundated and natural conditions was made by using the average *M_DMT_* values within the zone of influence of the foundation element, considered equal to 2*B* (*B* = 0.30 m). The *M_DMT_* values calculated from the DMT data are in good agreement with the other two techniques (oedometer and plate load tests), as shown in Figure 10. This is a good indicator that the DMT is an appropriate in situ testing approach for estimating soil stiffness under working loading and consequently can be used for an adequate foundation settlement prediction. Figure 10 shows a greater difference between the constrained modulus values determined by different techniques at 4.0 m depth, which was also observed in the prediction of the load–settlement curve at this depth.

It is important to note that the differences observed between the values from the PLT and DMT to those measured in the oedometer tests (Figure 10), especially for 3.0 and 4.0 m depth, may be related to the soil sampling process, as well as to the effect of the unsaturated soil condition.

The inundation process reduced the constrained modulus along the entire soil profile. This behaviour is associated with substantial suction reduction caused by increasing the soil moisture content, and the performance of shallow foundations on unsaturated tropical soils is strongly influenced by the soil suction [24,27,28]. Table 3 indicates that, overall, *M_DMT_* had an average reduction of 56% from natural to inundated conditions. This indicates that the *M* modulus determined by these three distinct tests is sensitive to variations in soil suction.

### 5.3. Load–Settlement Curves

Figure 11 shows the load–settlement curves obtained from in situ plate load tests (PLT) and predicted from the DMT, as presented in the Item 2.2, for both natural and inundated conditions. It can be observed in this figure that the curves predicted from DMT are in good agreement with those obtained by PLT.

Soil inundation eliminates the effect of the seasonal variability in suction and may be one of the reasons why the curves for the inundated tests show better agreement. The comparison between predicted (DMT) and measured (PLT) curves should be performed with caution for the natural soil condition, since these are susceptible to the effect of suction and may present seasonal variability along the active zone, as discussed by Giacheti et al. [36]. The active zone is commonly defined as the region of water content and soil suction change seasonally due to climatic variations [44,45].

Shallow foundation settlement prediction methods can be assessed by analysing their accuracy when comparing the measured and predicted settlements [46,47]. Figure 12 shows the differences between the settlements predicted by DMT for the working condition (*s*/*B_eq_* = 1.8%, i.e., for 4.78 mm for a plate with 300 mm diameter) and those obtained by PLT for both inundated and natural conditions. In addition, Table 4 summarizes the predicted settlement values by PLT at the working conditions. The settlements for natural conditions were 2.5, 3.8, 4.6, and 6.9 mm, respectively, for 1.0, 2.0, 3.0, and 4.0 m depth. The settlements for inundated condition were 2.1, 2.9, 3.6, and 5.3 mm, respectively, for 1.0, 2.0, 3.0, and 4.0 m depth. Good agreements were observed, since most of the data points are within the satisfactory range, as discussed by Monaco et al. [7].

## 6. Conclusions

This study presents an approach that uses a simple device attached to the dilatometer blade to perform local inundation of the soil during DMT. This device allows determining the in situ constrained modulus (*M_DMT_*) for the natural and inundated conditions and predicting the load–settlement curves by DMT for both conditions. The curves predicted by DMT and measured by PLT were strongly influenced by soil inundation at all studied depths. The results showed good agreement between the curves predicted by DMT and those obtained by PLT for natural and inundated conditions, and this approach proved to be an adequate method for predicting curves under different in situ conditions. It was observed for all investigated depths that the *M_DMT_* was influenced by soil suction and had an average reduction of 56% from natural to inundated conditions, which implies that for predicting load–settlement behaviour on collapsible soils, seasonal variability should be taken into account. The effect of the unsaturated condition in the design stage of collapsible soils can be a problem if not properly considered. Obtaining parameters from test data carried out during a dry period can lead to typical problems of unsaturated soils in case of increasing soil moisture content, such as the collapsibility phenomenon. On the other hand, obtaining parameters via inundated tests can lead to highly conservative practices, where parameters for saturated soils are assumed, even for regions where the unsaturated soil behaviour prevails.

The local inundation using the device attached to the DMT blade provided a simpler in situ investigation method to obtain parameters for the inundated condition, which can be a practical approach for shallow foundation design, mainly in collapsible soils, since it improves the settlement prediction for both natural and inundated conditions and depends only on DMT data with little correlation dependency.

The authors suggest that future works combine laboratory and in situ tests as well as numerical modelling [48,49] to better understand the effect of soil suction and its variability over the year on load–settlement curves of shallow foundations.

## Figures and Tables

**Figure 1 sensors-23-01429-f001:**
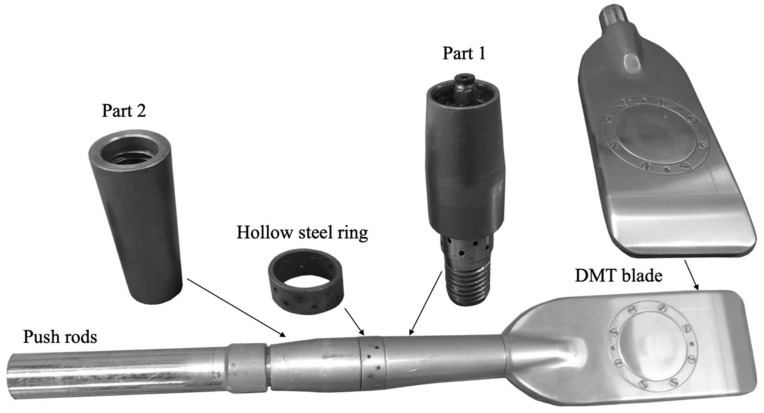
Parts of inundation device and system attached to the DMT blade. Adapted from [32].

**Figure 2 sensors-23-01429-f002:**
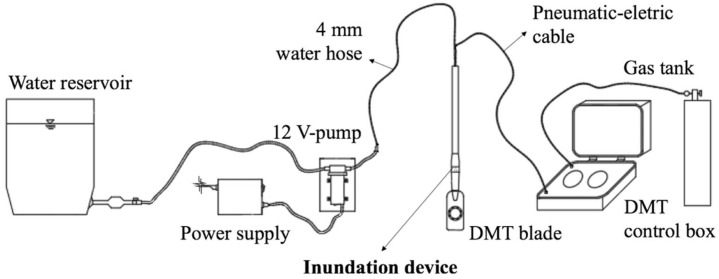
Schematic representation of the DMT test with the inundation device. Adapted from [32].

**Figure 3 sensors-23-01429-f003:**
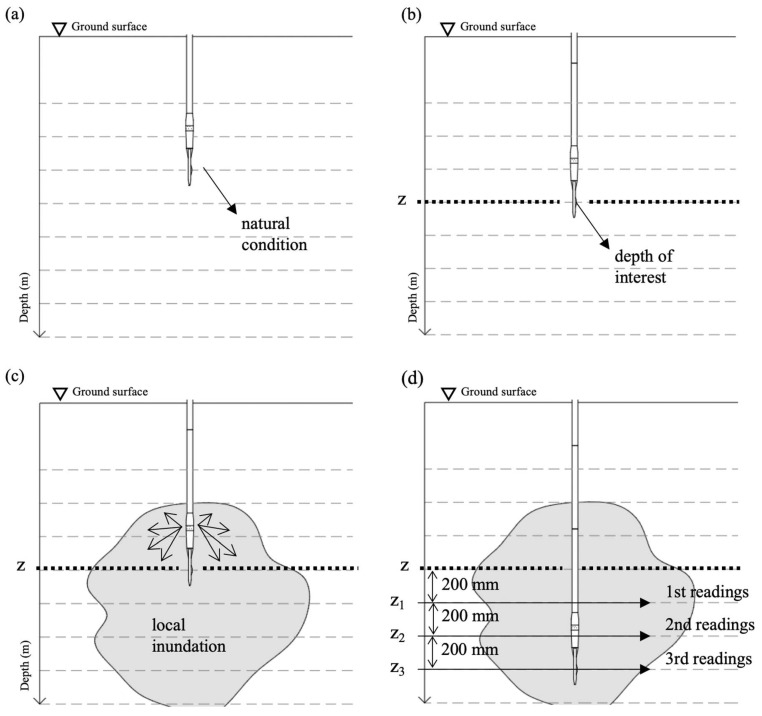
Schematic representation of DMT inundation process, (**a**) test under natural condition, (**b**) depth of interest; (**c**) local inundation and (**d**) readings under inundated condition. Adapted from [32].

**Figure 4 sensors-23-01429-f004:**
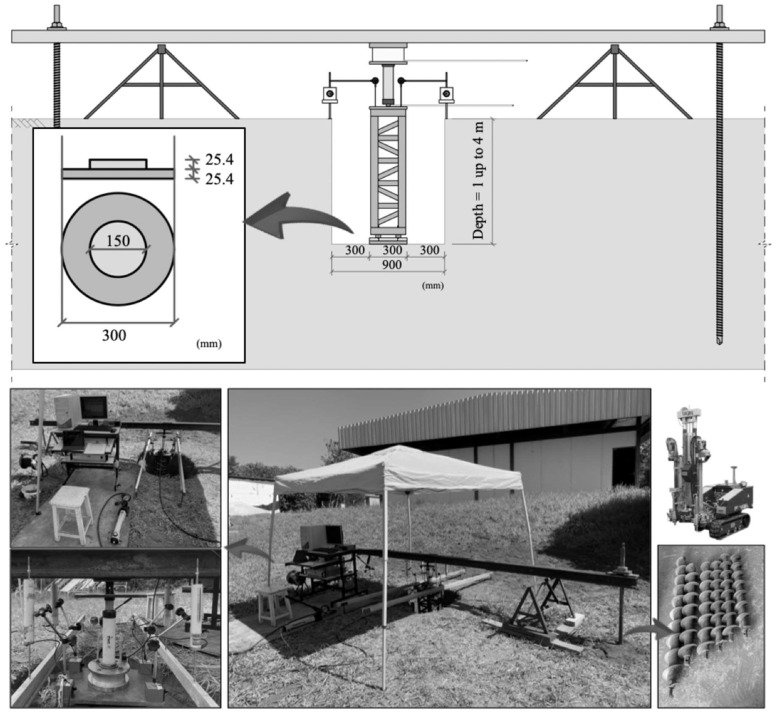
Equipment and setup for the plate load tests carried out at the site.

**Figure 5 sensors-23-01429-f005:**
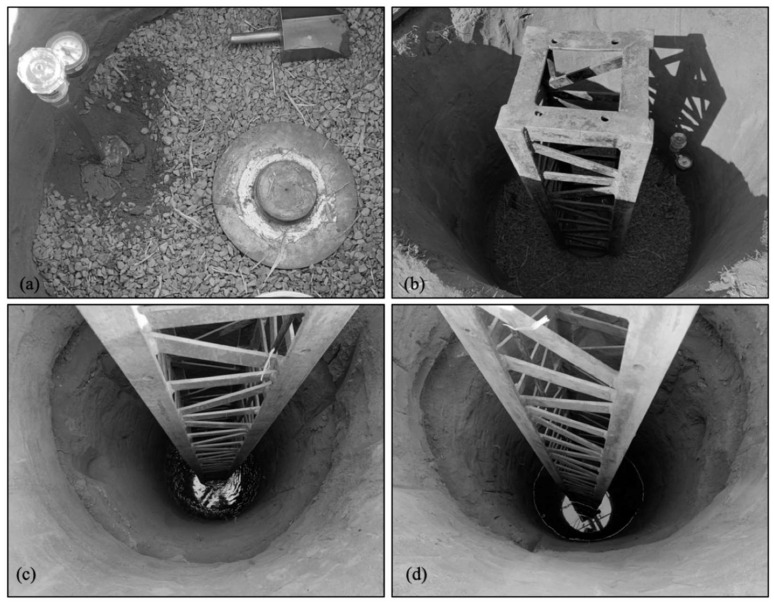
(**a**) Base of the pit prior to the inundation, (**b**) test assembled prior to the inundation, (**c**) process of inundation and (**d**) PLT carried out under inundated condition.

**Figure 6 sensors-23-01429-f006:**
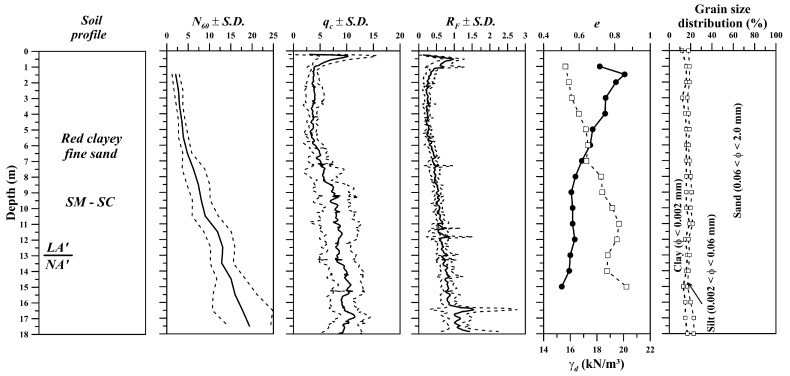
In situ and laboratory tests previously carried out in the experimental site. Adapted from [38,39].

**Figure 7 sensors-23-01429-f007:**
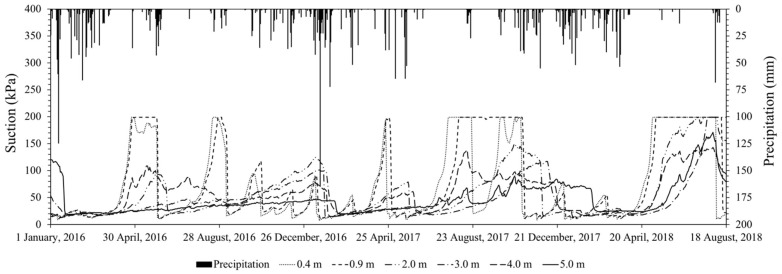
Suction monitoring by granular matrix sensors and precipitation data.

**Figure 8 sensors-23-01429-f008:**
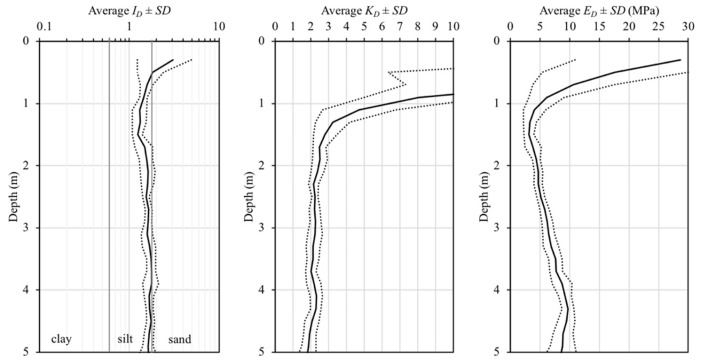
Average profile as well as plus and minus one standard deviation (*SD*) of *I_D_*, *K_D_*, and *E_D_* determined in natural conditions.

**Figure 9 sensors-23-01429-f009:**
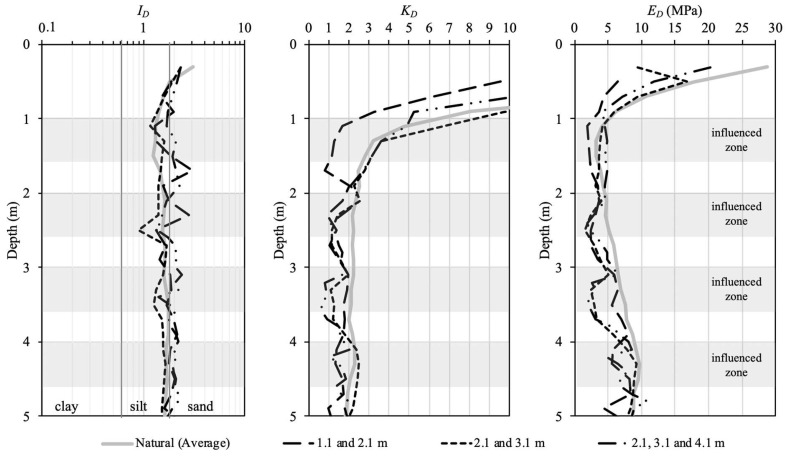
*I_D_*, *K_D_*, and *E_D_* profiles obtained after local inundation and average profile for the natural condition.

**Figure 10 sensors-23-01429-f010:**
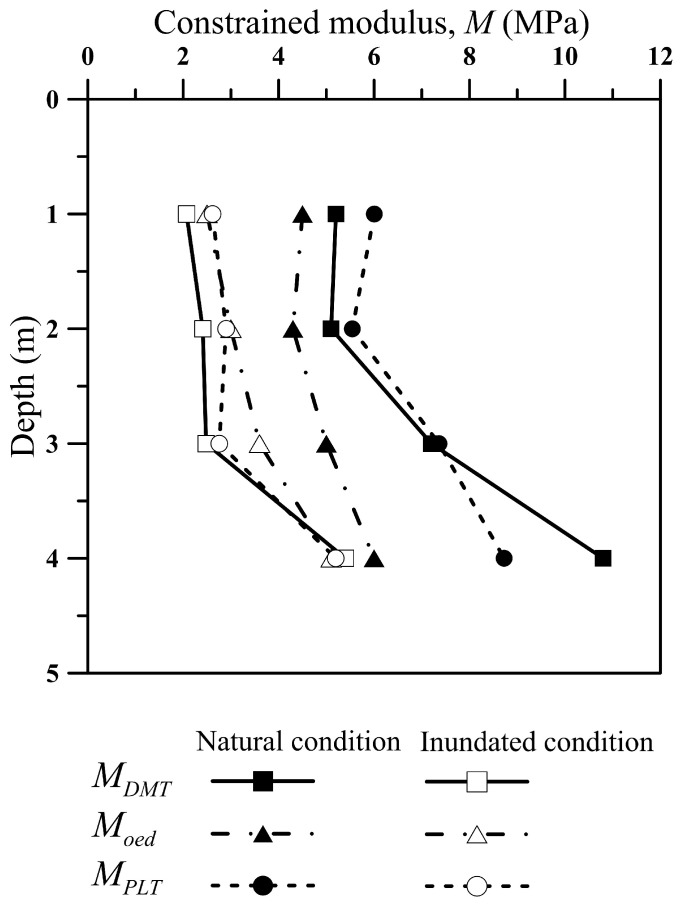
Constrained modulus values determined from the DMT, oedometer test and PLT, under natural and inundated condition for the study site.

**Figure 11 sensors-23-01429-f011:**
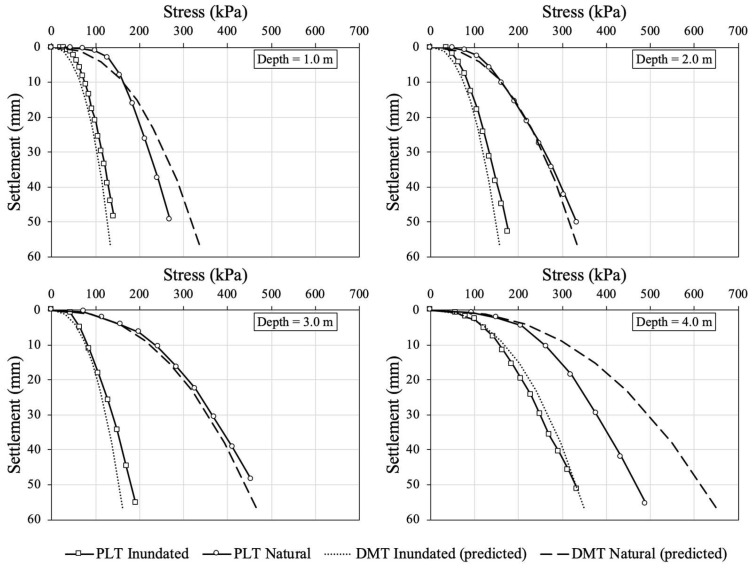
Load–settlement curves measured by PLT and predicted by DMT results, under natural and inundated conditions, for 1.0, 2.0, 3.0, and 4.0 m depth.

**Figure 12 sensors-23-01429-f012:**
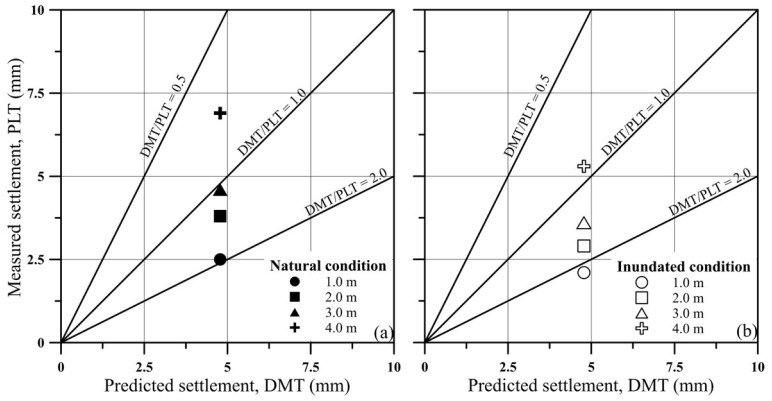
Settlements estimated from DMT vs. those obtained by PLT for the natural (**a**) and inundated (**b**) conditions of the study site.

**Table 1 sensors-23-01429-t001:** Determination of *R_M_* correction factor [10].

*I_D_* ≤ 0.6	*R_M_* = 0.14 + 2.36 log *K_D_*
*I_D_* ≥ 3.0	*R_M_* = 0.50 + 2.0 log *K_D_*
0.6 < *I_D_* < 3.0	*R_M_* = *R_M_*_,0_ + (2.5 − *R_M_*_,0_) log *K_D_*, with: *R_M_*_,0_ = 0.14 + 0.15 (*I_D_* − 0.6)
*K_D_* > 10.0	*R_M_* = 0.32 + 2.18 log *K_D_*
*R_M_* < 0.85	set *R_M_* = 0.85

**Table 2 sensors-23-01429-t002:** Intermediate parameters obtained for both natural and inundated conditions along the influenced zone (0.60 m).

Depth (m)	Natural				Inundated			
	*I_D_*	*K_D_*	*E_D_* (MPa)	*M_DMT_* (MPa)	*I_D_*	*K_D_*	*E_D_* (MPa)	*M_DMT_* (MPa)
1.1	1.31	4.71	4.05	7.13	1.52	1.66	1.88	1.63
1.3	1.32	3.22	3.30	4.57	1.88	1.25	2.15	1.83
1.5	1.24	2.80	3.13	3.89	2.81	1.13	3.24	2.76
2.1	1.63	2.38	4.74	5.30	1.97	1.15	2.65	2.25
2.3	1.60	2.15	4.67	4.75	1.55	1.26	2.58	2.19
2.5	1.54	2.23	5.09	5.33	1.95	1.21	3.29	2.80
3.1	1.59	2.23	6.45	6.80	1.86	0.91	2.72	2.31
3.3	1.68	2.11	6.84	6.91	1.75	1.06	3.01	2.56
3.5	1.75	2.12	7.61	7.75	1.78	0.94	3.02	2.57
4.1	1.66	2.31	9.21	10.07	2.17	1.02	4.77	4.06
4.3	1.69	2.28	9.71	10.52	1.95	1.63	7.14	6.12
4.5	1.74	2.06	9.45	9.38	2.07	1.61	6.80	6.02

**Table 3 sensors-23-01429-t003:** *M_DMT_* values for the natural and inundated condition and the reduction observed on this parameter.

Depth (m)	Natural *M_DMT_* (MPa)	Inundated *M_DMT_* (MPa)	Reduction (%)
1	5.21	2.07	60
2	5.16	2.41	53
3	7.20	2.48	66
4	10.05	5.40	46

**Table 4 sensors-23-01429-t004:** Measured settlement from PLT in working conditions.

Depth	Natural		Inundated	
(m)	Stress (kPa)	Settlement (mm)	Stress (kPa)	Settlement (mm)
1	119.4	2.5	47.5	2.1
2	118.2	3.8	55.3	2.9
3	165.0	4.6	56.9	3.6
4	230.2	6.9	123.7	5.3

## Data Availability

The datasets generated and analysed during the current study are available from the corresponding authors on reasonable request.

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
