# Peer review of "Suction Influence on Load–Settlement Curves Predicted by DMT in a Collapsible Sandy Soil"

_sensors, 2023, doi:10.3390/s23031429_

Round 1

Reviewer 1 Report

MAJOR REVISION

I am mainly concerned about the contribution of this work. It seems that you have described two test methods, namely, DMT and PLT, followed by data processing of the site results. I did not see some insightful discussions from the conclusion part.

(1) A keyword in this work is the "suction influence". Could you briefly summarize the impact of suction on the load-settlement curve?

(2) One suggestion on the reference format: it is not good to just use [x] as the object (e.g., line 59, etc.) or subject (e.g., line 63, line 97, etc.) in one sentence. You may combine with the authors' last name, i.e., Smith et al., [x] or Smith and Jordan [x].

(3) A lot of symbols and abbreviations appeared in this experimental paper, and I would suggest adding a nomenclature.

(4) What is the symbol "s" in Eq. (5)? If it implies the settlement, then the conventional failure stress will also be a function of "s". Is that correct?

(5) From Figs. 8 to 12, where do you apply Eqs. (1) to (6)?

(6) In Figs. 11 and 12, there are some estimated or predicted curves or points. How do you obtain these results? I cannot find any related mathematics.

(7) Please add some comments on future work. In my opinion, you may consider numerical simulation. For example, the strip load problem is somehow similar to PLT, thus you may consider hydromechanical coupling in the future, see DOI: 10.1016/j.compgeo.2022.104728 and include it if okay.

Zhang Q, Yan X, Li Z (2022) A mathematical framework for multiphase poromechanics in multiple porosity media. Computers and Geotechnics 146:104728

Author Response

Please see te attachment.

Reviewer 2 Report

The authors provide an interesting approach to predict the e load‒settlement curves, which is more convenient and cheaper than the PLT. In general, the manuscript is well written and easy to understand. However, the reviewer thinks the presentation of this manuscript needs to be improved before the publication. And the limitation of this method should also be pointed out in the paper. Major revision is recommended.

The authors are advised to incorporate following comments in the revision.

1、   L24-25The meaning of this sentence is unclear. Do the results predicted by DMT correspond to natural condition, and those by PLT correspond to inundated condition? Or it means that the predicted results under two conditions all agrees well? The same problem is in L68-70.

2、   L82: Please unify the type of reference. like “(Marchetti et al., 2001).”

3、   L103: Why the zone of influence can be considered as 2B. Please show more explanations.

4、   L151-152: Is the number of 300mm related to 200mm*3? Some detailed explanation is suggested to add in this part.

5、   The parameters chosen in the DMTs should be illustrated in the paper. Attached as a table.

6、   Please recheck the label in Fig.12, there are two lines represent the “DMT/PLT=0.5”?

7、   According to Fig.11, there exit some differences between the predicted results and PLT results, especially for the condition with the depth of 4m.

8、   Conclusion: the logic in this part is not clear, some contents with same meaning repeated several times, not concise enough.

9、   The predicted results show good performance for the soil in this paper. But, how to choose the empirical parameters for other types of soil? Some suggestions should be given in the paper.

Author Response

Please see te attachment.

Reviewer 3 Report

This study presents a device that can be attached to flat dilatometer tests (DMTs) to allow local inundation of the soil as part of the in-situ test campaign. The study is interesting and has good practical value. However, there are still some issues needed to improve to enhance the quality of the manuscript before publication. They are:

(1) The authors pointed out that the results from DMTs were in good agreement with the results of the plate load test. However, the testing resulting was just similar in the trend of change. How the authors explain those differences and which was the best? How to explain the reliability of the proposed DMTs?

(2) There are too many keywords in the manuscript. Please control them within 5.

(3) Please introduce the experimental results in the abstract to support the conclusion.

(4) The discrimination of lines under the three DMT parameters in Figure 9 was not clear to read. Please improve.

(5) In Figure 11, the results of Natural PLT and DMT (predicted) at 4m depth were relatively different from those at other depths. Please explain the reasons for this difference.

(6) The conclusion was too simple. Please add the detailed research results.

(7) Lines 119-127, the author should describe the relevant parameters of the equipment.

(8) Line 132, please explain how to inject water into the soil, whether it is a one-time injection. Besides, how about the injection speed?

(9) Lines 174-178, the description of the anchor bolt layout of the reaction system is too simple. Please show the layout of the anchor bolt.

(10) Please describe in detail the function of each submerged device in Figure 2.

(11) Please complete the information in Figure 6.

(12) Changes in the DMT intermediate parameters have a direct effect on the estimate of the deformability soil parameters However, this is not obvious in Figure 9.

(13) Please improve the citation of reference in the manuscript following the requirement of the journal.

Author Response

Please see te attachment.

Reviewer 4 Report

This manuscript discussed the method using DMT to predict load-settlement curves. I think the technical content of this paper is good. But there are still some concerns need the authors to address:

- Please clarify how the authors can ensure the suction data in Fig. 7 is measured accurately. Soil suction is very sensitive to measure, especially at the site. So it need a lot of controls.

- The introduction of this manuscript needed to be extended, as the discussions on the previous studies on load-settlement prediction models were quite limited.

- I also have questions about the novelty of this work. The authors just used a set of know calculations predict the load-settlement curves. So what is the key contribution of this work to the current literature? Please clarify it.

Author Response

Please see te attachment.

Round 2

Reviewer 1 Report

The revision is good.

Reviewer 3 Report

The authors have revised their manuscript following the comments, and can be accepted.